# Fluoroalkyl Pentacarbonylmanganese(I) Complexes as Initiators for the Radical (co)Polymerization of Fluoromonomers

**DOI:** 10.3390/polym12020384

**Published:** 2020-02-08

**Authors:** Roberto Morales-Cerrada, Vincent Ladmiral, Florence Gayet, Christophe Fliedel, Rinaldo Poli, Bruno Améduri

**Affiliations:** 1Institut Charles Gerhardt Montpellier, University of Montpellier, CNRS, ENSCM, Place Eugène Bataillon, 34095 Montpellier CEDEX 5, France; roberto.morales-cerrada@enscm.fr (R.M.-C.); vincent.ladmiral@enscm.fr (V.L.); 2Laboratoire de Chimie de Coordination (LCC), Université de Toulouse, CNRS, UPS, INPT, 205 route de Narbonne, BP 44099, 31077 Toulouse CEDEX 4, France; florence.gayet@ensiacet.fr (F.G.); christophe.fliedel@lcc-toulouse.fr (C.F.)

**Keywords:** fluoropolymers, manganese complexes, nuclear magnetic resonance, radicals, organometallic-mediated radical polymerization, vinylidene fluoride

## Abstract

The use of [Mn(R_F_)(CO)_5_] (R_F_ = CF_3_, CHF_2_, CH_2_CF_3,_ COCF_2_CH_3_) to initiate the radical polymerization of vinylidene fluoride (F_2_C=CH_2_, VDF) and the radical alternating copolymerization of vinyl acetate (CH_2_=CHOOCCH_3_, VAc) with *tert*-butyl 2-(trifluoromethyl)acrylate (MAF-TBE) by generating primary R_F_^•^ radicals is presented. Three different initiating methods with [Mn(CF_3_)(CO)_5_] (thermal at ca. 100 °C, visible light and UV irradiations) are described and compared. Fair (60%) to satisfactory (74%) polyvinylidene fluoride (PVDF) yields were obtained from the visible light and UV activations, respectively. Molar masses of PVDF reaching 53,000 g·mol^−1^ were produced from the visible light initiation after 4 h. However, the use of [Mn(CHF_2_)(CO)_5_] and [Mn(CH_2_CF_3_)(CO)_5_] as radical initiators produced PVDF in a very low yield (0 to 7%) by both thermal and photochemical initiations, while [Mn(COCF_2_CH_3_)(CO)_5_] led to the formation of PVDF in a moderate yield (7% to 23%). Nevertheless, complexes [Mn(CH_2_CF_3_)(CO)_5_] and [Mn(COCHF_2_)(CO)_5_] efficiently initiated the alternating VAc/MAF-TBE copolymerization. All synthesized polymers were characterized by ^1^H and ^19^F NMR spectroscopy, which proves the formation of the expected PVDF or poly(VAc-*alt*-MAF-TBE) and showing the chaining defects and the end-groups in the case of PVDF. The kinetics of VDF homopolymerization showed a linear ln[M]_0_/[M] versus time relationship, but a decrease of molar masses vs. VDF conversion was noted in all cases, which shows the absence of control. These PVDFs were rather thermally stable in air (up to 410 °C), especially for those having the highest molar masses. The melting points ranged from 164 to 175 °C while the degree of crystallinity varied from 44% to 53%.

## 1. Introduction

Polyvinylidene fluoride (PVDF) [1,2,3,4,5] is the most produced fluoropolymer after polytetrafluoroethylene (PTFE). This specialty polymer has many relevant properties such as chemical inertness to strong acid and apolar solvents, ferroelectric, pyroelectric, and piezoelectric (when stretched) properties, as well as thermal and electrochemical resistances. It can be involved in many applications such as a binder and separator for lithium ion batteries, paints and coatings, piezoelectrical devices, water purification membranes, and more.

PVDF is usually prepared by radical polymerization of vinylidene fluoride (VDF), which is known to proceed mainly by head-to-tail monomer additions to afford “head” radicals, PVDF_H_^•^. The occasional (ca. 3%–5%) head-to-head additions to afford “tail” radicals, PVDF_T_^•^, are immediately followed by a tail-to-tail addition to regenerate PVDF_H_^•^ [6,7,8]. Reversible-deactivation radical polymerization (RDRP) of VDF has been possible via iodine transfer polymerization (ITP) [3,9,10,11,12,13] and, more recently, by reversible addition−fragmentation chain-transfer (RAFT) polymerization [14,15] and by organometallic-mediated radical polymerization (OMRP) [16,17]. Specifically, OMRP is based on the reversible trapping of the growing radical chain by a transition metal complex to generate an organometallic dormant species [18,19,20,21]. ITP and RAFT led to a loss of control once the inverted monomer additions led to the full conversion of the dormant species to the tail isomer, respectively PVDF_T_-I and PVDF_T_-xanthate, which are less easily reactivated than the corresponding PVDF_H_-I and PVDF_H_-xanthate species. Conversely, the OMRP technique does not suffer from this problem when the controlling complex is [Co(acac)_2_] because both dormant species are reactivated at similar rates [16], which is in agreement with predictions based on the density functional theory (DFT) calculations of the homolytic bond strengths [22].

Asandei et al. have demonstrated the utility of [Mn_2_(CO)_10_] in ITP in the presence of visible light irradiation because the photolytically produced [Mn(CO)_5_^•^] radicals are able to abstract the iodine atom from the less reactive PVDF_T_-I dormant species [3,13]. There is a question, however, about the possible direct trapping of the PVDF_H_^•^ and PVDF_T_^•^ chains to form (CO)_5_Mn-capped organometallic dormant chains and about the possible reactivation of such intermediates. Although the (CO)_5_Mn-PVDF bonds are predicted to be considerably stronger than the (acac)_2_Co-PVDF ones [22], their formation and subsequent thermal and/or photochemical reactivation may constitute an additional mechanism for controlling the polymerization, i.e., by OMRP. Therefore, we have recently embarked in synthetic and experimental bond dissociation enthalpy (BDE) determinations for a series of [Mn(R_F_)(CO)_5_] complexes where R_F_ stands for CF_3_ (**1**), CHF_2_ (**2**), CH_2_CF_3_ (**3**), and CF_2_CH_3_ (**4**) with the purposes of (*i*) providing an experimental benchmark to validate the DFT calculations and (*ii*) testing the possible action of these complexes as initiators in OMRP [23]. This investigation has revealed that the complex with the highest Mn-R_F_ BDE is **1** for which the kinetic activation enthalpy is Δ*H*_exp_ = 53.8 ± 3.5 kcal·mol^−1^. This value is very close to the DFT-calculated one (55.1 kcal·mol^−1^). The other determined values are also in line with the DFT predictions (for **2**: Δ*H*_exp_ = 46.3 ± 1.6 vs. Δ*H*_DFT_ = 48.0 kcal·mol^−1^; for **3**: Δ*H*_exp_ = 50.6 ± 0.8 vs. Δ*H*_DFT_ = 50.5 kcal·mol^−1^). Compound **4** could not be isolated in a sufficiently pure state for an experimental measurement of the BDE, whereas the DFT calculations suggest that it should contain a homolytically weaker bond (46.0 kcal·mol^−1^) [22]. However, its acyl precursor, [Mn(COCF_2_CH_3_)(CO)_5_] (**5**), could be obtained as a pure crystalline solid [23]. The inability to isolate **4** in a pure form may be related to the weaker Mn-CF_2_CH_3_ bond. All alkyl compounds were obtained by thermal decarbonylation of the corresponding acyl derivatives. However, while compounds **1**, **2,** and **3** are sufficiently robust and do not decompose during the decarbonylation step, compound **4** is more fragile. However, acyl derivative **5** is a suitable precursor for the generation of **4**, and, thus, of CH_3_CF_2_^•^ radicals in situ. Preliminary polymerization tests have confirmed that the Mn-CF_3_ complex (**1**) is able to initiate the VDF polymerization under both thermal and photochemical (visible, UV) conditions [23]. This article illustrates in fuller details the ability of complexes **1**, **2**, **3,** and **5** to initiate and attempt controlling the polymerization of VDF and other monomers under thermal and photochemical conditions.

## 2. Materials and Methods 

### 2.1. Materials 

Vinylidene fluoride (VDF) and *tert*-butyl 2-(trifluoromethyl)acrylate (MAF-TBE) were kindly supplied by Arkema and Tosoh Finechem Corp., respectively, and used as received. Complexes **1**, **2**, **3**, **5,** and **6** were synthesized as described in a previous contribution [23]. Dimethyl carbonate (DMC, ≥99%, Merck KGaA), acetone (VMR Chemicals), pentane (VMR Chemicals), dimethyl formamide (VMR Chemicals), benzene-*d_6_* (99.5%D, Eurisotop), acetone-*d_6_* (99.5%D, Eurisotop), DMSO-*d_6_* (99.5%D, Eurisotop) and DMF-*d_7_* (99.5%D, Eurisotop) were used as received.

### 2.2. Instrumentation

The Nuclear Magnetic Resonance (NMR) spectra were recorded on a Bruker Avance III HD 400 MHz spectrometer. The instrumental parameters were as follows. ^1^H NMR: flip angle = 30°, acquisition time = 5.7 s, pulse delay = 2 s, number of scans = 64, pulse width = 3.05 μs. ^19^F NMR: flip angle = 30°, acquisition time = 2 s, pulse delay = 2 s, number of scans = 64, and pulse width = 3.76 μs. The probe has lower background ^19^F signals compared to standard dual-channel probes. Size- Exclusion Chromatography (SEC) with 0.1 M LiBr/DMF as the eluent, calibrated with poly(methyl methacrylate) (PMMA) standards from Polymer Laboratories, was run with a Varian Prostar (model 210) pump at a flow rate of 0.8 mL·min^−1^ using two 300-mm long, mixed-D PL-gel and 5-μm columns (molar mass range: 2·10^2^–4·10^5^ g·mol^−1^ from Polymer Laboratories) thermo-stated at 70 °C, connected to a Shodex (model RI-101) refractometer detector. The sample concentration was ca. 10 mg·mL^−1^. The thermogravimetric analyses (TGA) of the purified and dried (co)polymer samples were performed in air using a TGA 51 apparatus from TA Instruments at a heating rate of 10 °C min^−1^ from room temperature to 580 °C. Differential scanning calorimetry (DSC) analyses of these samples were carried out using a Netzsch DSC 200 F3 instrument under an N_2_ atmosphere. The DSC instrument was calibrated with noble metals and checked before analysis with an indium sample (*T_m_* = 156.6 °C). After its insertion into the DSC apparatus, the sample was initially stabilized at 20 °C for 10 min. Then, the first scan was made at a heating rate of 10 °C·min^−1^ up to 200 °C. It was then cooled to −50 °C. Lastly, a second scan was performed at a heating rate of 10 °C·min^−1^ up to 200 °C. Melting points (*T_m_*s) were evaluated from the second heating, taken at the maximum of the enthalpy peaks, and its area determined the melting enthalpy (Δ*H_m_*). This ensured the elimination of the thermal history of PVDFs during the first heating. The degrees of crystallinity of PVDFs were determined using Equation (1).
Degree of crystallinity (χ) = (∆*H_m_*/∆*H_c_*) × 100(1)
where ∆*H_c_* (104.5 J g^−1^) corresponds to the enthalpy of melting of a 100% crystalline PVDF [24] while ∆*H_m_* is the heat of fusion (determined by DSC in J g^−1^), respectively.

### 2.3. VDF Polymerizations with [Mn(R_F_)(CO)_5_]

The radical polymerizations of VDF were carried out in thick-walled 12 mL Carius tubes. As a representative example, a solution of [Mn(CF_3_)(CO)_5_] (128 mg, 0.48 mmol) in 5 mL of dimethyl carbonate was introduced and degassed by three freeze-pump-thaw cycles. Then, the tube was cooled to the liquid nitrogen temperature and gaseous VDF monomer (1.5 g, 23.4 mmol) was transferred using a custom-made manifold that enables the accurate measurement of the gas quantity (using a “pressure drop vs. mass of monomer” calibration curve). The tube was then sealed under static vacuum at the liquid nitrogen temperature. For the thermal activation experiments, the tubes were placed horizontally in a thermostatic shaking water bath. For the visible light activation experiments, the tubes were placed horizontally in a tube roller shaker with 3 LED bulbs (Diall 1102270698, 14 W, 1521 Lm), irradiated from above, and placed 2 cm from the tubes. The proximity of the bulbs caused the tubes to warm up to 40 °C. For the UV activation experiments, the tubes were equipped with a small magnetic stirring bar and the solutions were stirred vertically in a Rayonet RPR-200 UV reactor equipped with sixteen 300-nm wavelength UV-lamps of 35 W each. Despite the fan placed inside the UV chamber, the heat generated by the UV lamps caused the tubes to warm up to 50 °C. After a certain time (ranging from 2 to 24 h), each tube was frozen into liquid nitrogen, opened, and the solvent was evaporated in a rotary evaporator. The monomer conversion was calculated from the PVDF yields (Equation (2), where m_PVDF_, m_VDF_, and m_complex_ stand for the recovered mass of PVDF, the initial mass of VDF, which was obtained from the pressure measurement, and the mass of complex **1** used to initiate the polymerization, respectively) since an accurate direct measurement of the VDF conversion is complicated by the gaseous state of the monomer.
Conversion (%) = 1 − [m_PVDF_/(m_VDF_ + m_complex_)] × 100(2)

The resulting polymers were dissolved in DMF-*d_7_*, DMSO-*d_6_*, or acetone-*d_6_* (depending on the solubility) and characterized by ^1^H and ^19^F NMR spectroscopy. Molar masses and dispersities were determined by SEC in DMF (refractive index, calibrated with PMMA standards).

### 2.4. Copolymerization of VAc and MAF-TBE

The radical co-polymerizations of VAc and MAF-TBE were carried out in a Schlenk tube under an argon atmosphere. As a representative example, 93.3 mg of [Mn(CO)_5_(CH_2_CF_3_)] (3.35·10^−1^ mmol) were introduced in a Schlenk tube. Then, 1.554 g of VAc (18.1 mmol) and 3.324 g of MAF-TBE (16.9 mmol) were added. The resulting solution was degassed by three freeze-pump-thaw cycles, and placed under argon. The solution was heated up to 80 °C for 18 h, and the resulting dark yellow solid was dissolved in 15 mL of acetone. The polymer was precipitated by adding the solution dropwise to 200 mL of cold *n*-pentane and then the precipitate was dried at 80 °C under vacuum for 16 h, which yielded 4.053 g of a yellow-brownish solid (83% yield). The polymer was dissolved in acetone-*d_6_* and analyzed by ^1^H and ^19^F NMR spectroscopies. The molar masses and dispersities were determined by SEC in DMF (refractive index, calibrated with PMMA standards).

## 3. Results

### 3.1. Radical Polymerization of VDF with **1** as an Initiator

Several polymerizations of VDF using complex **1** as the initiator were carried out under thermal and photochemical conditions (Scheme 1). Among all the fluoroalkylpentacarbonylmanganese(I) complexes synthesized in previous contributions [23,25], compound **1** possesses the strongest Mn-C BDE, according to the computational and experimental investigations data. Therefore, it is necessary to apply a relatively high temperature to cleave the Mn-CF_3_ bond. In the case of thermal activation, the experiment was run at 100 °C, which is a temperature at which the determined half-life in C_6_D_6_ is 37.2 minutes and considers that 99% of the complex decomposes (t_99%_) after 4.1 h [23]. Once the F_3_C^•^ radicals have been generated, their addition to a monomer should be fast. The photochemical initiation was carried out either by visible light irradiation using LED bulbs or by UV irradiation using a monochromatic 300 nm lamp.

Control experiments for both thermal and photochemical (visible light) activations carried out in the presence of [Mn_2_(CO)_10_] instead of **1** did not yield to any PVDF. Since [Mn_2_(CO)_10_] is known to generate [(CO)_5_Mn^•^] radicals under both thermal and photochemical conditions [26], it can be concluded that [(CO)_5_Mn^•^] is not capable of adding to VDF to initiate the polymerization. This result confirms previous studies reported by Bamford et al. [27] and by Asandei et al. [13], which showed that [(CO)_5_Mn^•^] photoproduced from [Mn_2_(CO)_10_] is unable to initiate the radical polymerization of VDF (although it does initiate those of C_2_F_4_ and C_2_F_3_Cl) [27].

Several polymerizations were carried out with complex **1** as an initiator under thermal and photochemical activations, which stopped the reactions at different times. The reaction conditions and analytical data of the resulting polymers are summarized in Table 1. All reactions were performed targeting a degree of polymerization of 50, expecting a molar mass of ca. 3500 g·mol^−1^ under the assumption of a well-controlled OMRP [28] with the pentacarbonylmanganese radical as a moderating agent.

A first thermal VDF polymerization test (with protection from light) was performed at 50 °C for 24 h (entry 1). Such conditions did not lead to the formation of any PVDF, which means that this temperature is too low to homolytically cleave the Mn-CF_3_ bond in **1**. Extrapolation of the previously determined activation parameters to 50 °C leads to a half-life of 9 × 10^8^ s (ca. 29 years) for compound **1** at this temperature [23]. However, PVDF was produced in high yields (up to 70% after 24 h) at higher temperature (100 °C, entries 2 to 7). As mentioned above, 100 °C leads to a relatively rapid generation of F_3_C^•^ radicals from **1** in C_6_D_6_ (t_½_ ≈ 37 min, t_99%_ ≈ 4 h) [23]. Hence, new polymer chains should be generated 4 h after the beginning of the polymerization assuming a similar t_1/2_ in DMC/VDF to that determined in C_6_D_6_.

The most interesting result concerning the thermal activation is that the polymer yield keeps increasing beyond 4 h of reaction and even after 18 h (e.g., from 49% after 18 h, entry 6, to 68% after 24 h, entry 7). This result indicates either the intervention of a reversible PVDF^•^ trapping by [Mn(CO)_5_^•^] (i.e. an OMRP mechanism) or a much longer initiator half-life in the DMC/VDF medium relative to C_6_D_6_. The dispersities of the recovered polymers continuously increased as a function of conversion, which is consistent with a relatively slow initiator decomposition, whereas the molar masses initially increased, and then decreased (entries 3–7). These characteristics indicate poor control, although the presence of some degree of OMRP trapping cannot be excluded.

The polymerization initiation was also very effective under photochemical conditions. Initiation by visible light led to PVDF in high yields (60% after 24 h, entry 11), as well as by UV irradiation (74% after 24 h, entry 14). It is worth noting that the measured reaction temperature, self-regulated by heat released by the LED bulbs, was 40 °C during the experiment, and 50 °C in the UV reactor. However, entry 1 proves that this temperature is not sufficient to release radicals. Thus, it is confirmed that visible light and UV irradiation lead to Mn-CF_3_ bond cleavage to produce F_3_C^•^ radicals. For these experiments, a tendency for a slight decrease of the polymer dispersities as a function of conversion was observed. However, visible light activation, contrary to the UV one, induces a decrease of the molar masses with conversion. Their values are much greater than expected based on a controlled polymerization. This may be caused by a lower initiation efficiency or slower Mn-CF_3_ homolytic cleavage under visible irradiation than under UV or at 100 °C. Thus, it appears that generation of the {[Mn(CO)_5_^•^], PVDF^•^} radical pair is not followed by efficient trapping to produce an organometallic dormant species. Rather, radicals escape from the solvent cage and subsequent uncontrolled chain growth and termination processes predominantly occur.

The polymerization kinetics are close to first-order and confirm that the polymerization proceeds with an approximately constant radical concentration (Figure 1, Appendix A and Appendix A).

The size exclusion chromatograms of PVDF samples obtained by thermal activation (Appendix A) exhibit two different populations: the main one displays a significant molar mass increase at the beginning (entries 2–3) and then a decrease (entries 3–7), which proves the lack of control during the reaction. The negative signal in the size exclusion chromatograms results from the lower refractive index of PVDF relative to the DMF eluent [5,29,30]. The signal of the second population at low molar mass can be attributed to oligomers arising from transfer reactions. Thus, the behavior of this polymerization is a conventional radical polymerization. This is also confirmed by an increase of the dispersity accompanied by a decrease of the average molar masses (Appendix A), as frequently observed in conventional radical polymerization. This behavior is also likely caused by hydrogen atom transfer from DMC, which stops the growing macroradicals to yield dead PVDF chains and generates new radicals that can create new chains initiated by the H_3_COC(O)OCH_2_^•^ radical. However, despite the lack of control, a high conversion (68%) was reached after 24 h, which establishes complex **1** as an efficient initiator of VDF polymerization.

The size exclusion chromatograms in Appendix A show the PVDF mass evolution during the visible light initiation. As above, the average molar mass decreases (compare entries 9 and 13). The plots of molar masses and dispersities versus conversion of the polymerization carried out under visible light activation (entries 9, 10, and 11, Table 1) are represented in Appendix A. Molar masses decrease with conversion such as in a conventional radical polymerization. Surprisingly, it should be noted that the dispersity slightly decreases with time.

The SEC traces of the crude polymers obtained by UV irradiation are represented in Appendix A. The evolution of the number of average molar masses and dispersities versus VDF conversion (Figure 2) clearly shows an increase of the molar mass and a decrease of dispersity during the polymerization. Both parameters evolve significantly at the beginning of the polymerization. This evolution seems less pronounced at the end of the reaction. In addition, the initially present low molar mass population gradually disappears. The observed trends in this polymerization are suggestive of a certain level control for an RDRP. However, the final polymer dispersity was relatively high (Ð = 1.63) while the M_n_ (26,000 g·mol^−1^) was much lower than that obtained from the visible light activation, and significantly higher than that observed with thermal initiation (Appendix A). In addition, the M_n_ growth and the dispersity drop were not linear, which suggests a certain control mainly at the beginning of the polymerization.

### 3.2. NMR Characterization of PVDFs

The ^1^H NMR spectra of the recovered polymers after 24 h for the three different initiation methods (Figure 3, Appendix A, Appendix A and Appendix A) exhibit the characteristic PVDF signals centered at 2.4 (minor) and 3.0 (major) ppm, assigned to the reverse -CF_2_C*H_2_*-C*H_2_*CF_2_- (tail-to-tail) and regular -CH_2_CF_2_-C*H_2_*CF_2_- monomer additions, respectively [3,6,13,31,32]. The triplet (^2^*J*_HF_ = 55 Hz) of triplets (^3^*J*_HH_ = 7 Hz) at δ 6.44 ppm is attributed to the proton in the -CH_2_-CF_2_*H* chain-ends generated by hydrogen transfer from the solvent, or by backbiting, which both involve the main (-CH_2_-CF_2_^•^) macroradical [33]. Similarly, this hydrogen transfer to the minor -CF_2_-CH_2_^•^ macroradical is responsible for the observed triplet (^3^*J*_HF_ = 18 Hz) centered at δ 1.85 ppm, which is assigned to the methyl end-group in -CF_2_-C*H_3_* [6,13,31]. From the relative signal integrals, it seems that the hydrogen transfer rates are lower for the polymerization carried out under visible light and UV light activation compared to the thermal initiation, according to the different measured molar masses (entries 11, 14, and 7 in Table 1, M_n_ = 40300, 26,000, and 16,900 g·mol^−1^, respectively). The transfer side-reaction leads to a decrease of the number of average molar mass and to an increase of dispersity, as observed in entries 3–7. In addition, two relatively intense resonances at δ 4.38 and 3.79 ppm in an approximate 2:3 ratio may be assigned to the –CF_2_-CH_2_-C*H*_2_-OC(O)O-C*H*_3_ chain ends, which results from a chain transfer to the DMC solvent [14] because their integrated intensities are consistent with that of the CH_2_CF_2_*H* signal. This is simultaneously generated by the same process (1:2:3 for the δ 6.44, 4.38, and 3.79 ppm resonances, respectively) [6,14].

The ^19^F NMR spectra of the PVDF obtained from the three activation methods (Appendix A) also exhibit all the expected resonances of PVDF [6,13,14,31,32,34,35]. The characteristic signal centered at −91.9 ppm (**a** in Figure 4) is assigned to the regular -CH_2_C*F_2_*-CH_2_CF_2_- (head-to-tail, H-T) sequences. The head-to-head (H-H, -CH_2_C*F_2_*-C*F_2_*CH_2_-) sequences, which are systematically followed by tail-to-tail (T-T, -CF_2_*CH_2_-CH_2_*CF_2_-) sequences, give rise to the resonances at −113.9 and −116.2 ppm (**c** and **d**, respectively). The percentage of H-H additions, obtained from the integrals of the ^19^F NMR signals (for the 24-hour-reaction, the fractions are 3.9%, 3.5% and 3.9% for the thermal, visible light and UV initiations, respectively) is typical for a PVDF produced by radical polymerization [36]. The F_3_C^•^ radicals generated from **1** should preferentially attack the VDF tail (CH_2_), as described in previous studies [32,34,37,38,39]. The head-end attack was reported to occur at less than 3% frequency at 150 °C [37]. Thus, the use of this radical should not significantly affect the percentage of inverted monomer additions. The CF_3_-CH_2_-CF_2_- chain-end produced by the initiation of the F_3_C^•^ radical could be shown by a small signal at δ –60.8 ppm, which confirms the previous studies [32,34,39], but only for the PVDF obtained by UV activation despite the high molar mass of the polymer. On the other hand, the expected signals for the chain ends generated by the transfer processes are always visible, as displayed in the spectra (**g** and **h** in Figure 4 and Appendix A, **h** and **i** in Appendix A). In addition, several unidentified small signals are present at δ −150.0, −99.0, −98.3, and −85.0 ppm in the case of the thermal activation, and an even smaller one at δ −85.0 ppm for the UV activation, which suggests that side-reactions occurred during the VDF polymerization carried out under these conditions. These signals were not observed for the polymers obtained by visible light initiation. It is important to underline that the ^19^F NMR resonance of **1** (expected at δ 5.65 ppm) [23] was not observed in any of the isolated PVDF products, which indicates full consumption of this compound.

### 3.3. Thermal Properties of the Resulting PVDFs

The thermal properties of the PVDF samples obtained by thermal or photochemical activations of compound **1** after a 24 h-reaction was studied by thermogravimetric analysis (TGA) and differential scanning calorimetry (DSC). Figure 5 displays the TGA thermograms of three PVDFs achieved from various initiations while Table 2 summarizes the temperatures at which 2%, 5%, and 10% weight losses are noted.

As shown in Table 2, the higher the molar mass of PVDF is, the higher its thermal stability is. The polymer synthesized under visible light, with the highest molar mass (M_n_ = 40,300 g mol^−1^), exhibits a 2% weight loss at the highest temperature (412 °C), whereas that obtained from thermal initiation, with the lowest molar mass (M_n_ = 16,900 g mol^−1^) shows the same weight loss at the lowest temperature (343 °C).

On the other hand, as expected, all DSC thermograms of the synthesized PVDFs (Figure 6) did not exhibit any glass transition (T_g_ is expected at −40 °C) [1,4]. However, the thermograms revealed the crystalline phase melting at higher temperatures for the higher molar masses (reaching 175 °C, which is in good agreement with previous reports [4] for the PVDF obtained by visible light initiation; see Table 3). The highest molar mass polymer also showed the highest degree of crystallinity (53%), whereas the lower molar mass polymers obtained by thermal or UV activations were less crystalline (44%).

### 3.4. Radical Polymerization of VDF with **2**, **3**, and **5** as Initiator

Compound [Mn(CHF_2_)(CO)_5_] (**2**) has a weaker Mn-C bond dissociation energy (BDE) than **1** and even weaker than [Mn(CH_2_CF_3_)(CO)_5_] (**3**), as confirmed by both the experimentally determined and the calculated BDEs [23]. Thus, the radical generation should require less energy compared to compound **1**. Therefore, 80 °C was chosen as the reaction temperature to initiate the polymerization by the thermal activation mode (t_½_ ≈ 46 min and t_99%_ ≈ 5 h for **2** at 80 °C in C_6_D_6_) [23]. This temperature was also selected for complex [Mn(COCF_2_CH_3_)(CO)_5_] (**5**), which is expected [23] to undergo thermal decarbonylation and to form the alkyl complex [Mn(CF_2_CH_3_)(CO)_5_] (**4**) in-situ before generating CH_3_CF_2_^•^ radicals. In the case of complex **3**, 90 °C was chosen as the reaction temperature due to its slightly higher BDE compared to complex 2 (t_½_ ≈ 21.6 min and t_99%_ ≈ 2.4 h at 90 °C in C_6_D_6_) [23]. The experimental conditions and the results are presented in Table 4.

The thermal activation of **2** at 80 °C for 24 h (entry 1 in Table 4) led only to a small amount of a brown-yellowish powder. However, this material does not contain any PVDF, as shown by the absence of observable resonances in the ^19^F NMR spectrum. Photochemical initiation, on the other hand, led to PVDF formation, which is only in very low yields (entries 2 and 3 of Table 4). This is much lower than for compound **1** under the same conditions. Similar low yields were obtained by thermally initiating the polymerization with complex **3** (entries 4–6). Photoactivation experiments were not carried out with this complex. In this case, the polymers were analyzed by SEC and found to have high molar masses, which is essentially independent of the monomer/initiator ratio and of the activation temperature (90 or 100 °C), which are quite similar to those of the polymers obtained in much higher yields using **1** as the initiator. However, these polymers surprisingly show low dispersities likely because of the very small quantities of the polymer that could be injected in the SEC apparatus (the signal/noise ratios were very close to unity in these cases). The polymerizations initiated by complex **5** led to slightly greater polymer yields, but still much lower than those initiated by complex **1**. Thermal activation gave only a 7% yield after 24 h (entry 7, Table 4), while visible light (entries 8–11) and UV light (entries 12–15) irradiation gave yields that increased with irradiation time up to 23% or 21%, respectively, after 24 h. The molar masses of these polymers, like those obtained from initiation with **1**, are high and the dispersities are moderate. Once again, this suggests the absence of a controlled chain growth of the type expected from OMRP. Since we have established that the Mn-R_F_ bond is homolytically weaker in compounds **2**, **3**, and **4** than in compound **1** [22,23], the lower polymer yield obtained with **2**, **3**, and **5**, at least under thermal activation conditions, cannot be attributed to a reduced primary radical flux. Rather, it appears that the CHF_2_^•^, CF_3_CH_2_^•^, and CH_3_CF_2_^•^ radicals (the latter being generated after decarbonylation of **5** to form **4** in situ) are not able to efficiently add to VDF.

The ^19^F NMR spectra of the recovered PVDF products display the usual features in terms of the monomer addition mode. Those obtained with initiation by photolytic activation of **2** (visible light, entry 2, Table 4, Appendix A, UV light, entry 3, Appendix A) show the presence of residual **2** in the polymer (resonance at δ −71.6 ppm), [23] particularly for the visible light irradiation experiment, which indicates a slow Mn-CHF_2_ bond photocleavage. Clearly, the efficiency of this process does not correlate with the BDE because the Mn-R_F_ bond is weaker for **2** than for **1**, whereas compound **1** is a quite efficient photo-initiator (vide supra). A relatively intense resonance at δ −114.4 ppm is assigned to the CHF_2_ chain-ends. Integration of this signal relative to the main PVDF resonance indicates that the obtained product is an oligomer with 8 monomer units. This result is similar for entries 2 and 3. This chain end likely corresponds to the primary radical produced by the initiator because there is no reason why the H atom transfer from DMC onto the PVDF_H_ macroradical, which leads to the same function, would be more frequent for the **2**-initiated polymerization than for that initiated by **1** (*cf.* with the intensity of the same resonance in Appendix A). In addition, two more resonances were detected in the sample and resulted from the visible light activation (Appendix A) at δ −75.9 and −74.7 ppm. These may correspond to a PVDF-CH = CF_2_ chain-end (*cf.* δ −72.2 and −73.4 ppm vs. acetone-*d*_6_ according to a previous report) [40].

For the PVDF generated by thermal initiation with compound **3**, no initiator resonances (e.g., the ^19^F NMR resonance, which is expected at δ −52.3 ppm) [23] were evident in the spectra of the isolated polymer (Appendix A). Thus, the initiator was fully activated under these conditions. No photoactivation experiments were carried out with this compound. The CF_3_CH_2_- chain end was highlighted by a small resonance at δ −60.8 ppm in the ^19^F NMR spectrum (Appendix A). Compound **5** was also fully activated under thermal conditions since ^19^F NMR resonances attributable to either **5** or to its decarbonylation product **4** (expected at δ −91.5 and −28.9 ppm, respectively) [23] were not detected (Appendix A).

### 3.5. Copolymerization of Vinyl Acetate with tert-Butyl 2-(Trifluoromethyl)acrylate Initiated by Complexes **3** and **6**

The aim of these investigations was to check the initiation capacity by using more reactive monomers of the complexes that do not efficiently initiate the VDF polymerization. For the purpose of this study, in addition to complex **3**, the acyl complex [Mn(COCHF_2_)(CO)_5_] (**6**), which generates complex **2** by thermal decarbonylation in situ was used instead of the isolated **2**. Four co-polymerizations of vinyl acetate (VAc) and *tert*-butyl 2-(trifluoromethyl)acrylate (MAF-TBE) were carried out using compounds **3** and **6** as initiators (Scheme 2). This copolymer was previously synthesized by conventional [41] OMRP using cobalt(II) acetylacetonate as a controlling agent [42] and RAFT polymerization [43]. All these four co-polymerizations led to copolymers, which display a perfectly alternating structure, in agreement with the measured reactivity ratios that are both close to zero [41]. The copolymerizations were carried out only with thermal initiation and the results are collected in Table 5.

The poly(VAc-*alt*-MAF-TBE) copolymer was obtained in moderate to high yields depending on the reaction temperature, which proves that compounds **3** and **6** can generate a flux of radicals able to initiate a radical polymerization. It is noted that the temperatures used for these polymerizations are equivalent or lower than that of the corresponding VDF polymerizations (i.e., up to 80 °C with compound **3** in entry 4, vs. 100 °C for the VDF polymerization). The ^1^H and ^19^F NMR spectra (Appendix A, respectively) of the isolated copolymer (Table 5, entry 4) are in agreement with the previously published ones [42,44] and clearly show the perfect alternation of comonomers since no ^1^H NMR signal centered at δ 4.9 ppm (the typical chemical shift of VAc-VAc dyads) was observed.

## 4. Discussion

The combined results of this investigation show that compound **1** is an efficient initiator for the polymerization of VDF under both thermal (100 °C) and photochemical (visible or UV light irradiation) conditions. This occurs by homolytic cleavage of the Mn-CF_3_ bond and generation of CF_3_^•^ radicals, which efficiently add onto the VDF tail (CH_2_), as reported in previous studies [32,34,37,38,39]. The head-end attack was reported to occur at less than 3% frequency at 150 °C [37]. On the other hand, the CHF_2_^•^ and CF_3_CH_2_^•^ radicals, generated from compounds **2** and **3**, respectively, and, to a certain extent, the CH_3_CF_2_^•^ radical released from compound **4** (which is, in turn, generated in situ from compound **5**) do not lead to efficient initiation of the VDF polymerization. A lower ability of compounds **2**, **3**, and **5** to produce enough radical flux can be excluded as a possible alternative explanation for the observed low PVDF yields (Table 4) because experimental and computational studies have shown that the Mn-R_F_ bond is homolytically stronger for **1**. Therefore, fluoroalkyl radicals can be generated from compounds **2**, **3**, and **5**, and are able to initiate the VDF polymerization. However, the yields of the resulting PVDF were rather low. Furthermore, the acyl precursor **6** (which yields **2** in situ by thermal decarbonylation) and **3** were shown to efficiently initiate the radical alternating copolymerization of VAc and MAF-TBE.

The effect of the radical and alkene substituents on the rate of the addition reaction has received some attention [45,46]. As pioneered by Tedder and Walton [37,45,46], Sokolov et al. [47] reported the reactivity of F_3_C^•^ and other perfluorinated radicals onto several fluorinated olefins and ethylene, concluding that the reactivity of F_3_C^•^ decreases with the number of fluorine atoms on the monomer (e.g., addition of F_3_C^•^ onto ethylene is almost 15 times faster than on VDF). However, to the best of our knowledge, there is no quantitative information on the rate of addition of differently substituted radicals such as CF_3_^•^, CHF_2_^•^, CF_3_CH_2_^•^, and CH_3_CF_2_^•^ onto the same monomer. The results of our studies were very unexpected because the PVDF-CH_2_CF_2_^•^ and PVDF-CF_2_CH_2_^•^ macroradicals can rapidly add onto the VDF molecule, preferentially to the tail end, during the VDF radical propagation step. We can, therefore, conclude that the rate of radical addition onto CH_2_ = CF_2_ is strongly affected by the radical substitution and, most importantly, that the CHF_2_^•^, CF_3_CH_2_^•^, and CH_3_CF_2_^•^ radicals cannot be considered as sufficiently good models for the reactivity of the PVDF head and tail macroradicals.

## 5. Conclusions

Fluoroalkylpentacarbonylmanganese(I) complexes are able to initiate the polymerization of VDF and the copolymerization of VAc and MAF-TBE by the homolytic bond cleavage of the Mn-R_F_ bond, which releases R_F_^•^ radicals. In the case of the VDF homopolymerization, the results are different depending on the nature of the R_F_ group and on the activation method, which yields PVDF in almost all cases but with different conversions, molar masses, and dispersities. The best results in terms of activation were achieved with [Mn(CF_3_)(CO)_5_] (**1**), with yields up to 74% from UV irradiation, 68% by thermal activation, or 60% by visible light activation. Although this complex has the strongest BDE among all investigated complexes [22,23], the resulting CF_3_^•^ radical is the most efficient one for the addition onto the VDF monomer. The second best result was surprisingly obtained with [Mn(COCF_2_CH_3_)(CO)_5_] (**5**), which generates the CH_3_CF_2_^•^ radical, following prior decarbonylation and [Mn(CF_2_CH_3_)(CO)_5_] (**4**) formation. Despite the fact that the latter complex has the lowest predicted BDE [22], it can initiate the polymerization of VDF in a relatively low yield under visible light and UV irradiation (21% and 23%, respectively). Nevertheless, this complex has a low efficiency in thermal activation at 80 °C. Complexes [Mn(CHF_2_)(CO)_5_] (**2**) and [Mn(CH_2_CF_3_)(CO)_5_] (**3**) are not efficient initiators under any of the tested activation conditions. However, these compounds are efficient initiators for the radical alternating copolymerization of VAc with MAF-TBE. Hence, the effect of the substituents on the radicals and their reactivity toward VDF are subtle. While the CF_3_^•^ radical is efficient, CHF_2_^•^, CF_3_CH_2_^•^, and CH_3_CF_2_^•^ radicals have shown a lower reactivity to be considered as suitable models for initiating the polymerization of VDF.

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
