# Peer review of "Fluoroalkyl Pentacarbonylmanganese(I) Complexes as Initiators for the Radical (co)Polymerization of Fluoromonomers"

_polymers, 2020, doi:10.3390/polym12020384_

Round 1

Reviewer 1 Report

The manuscript described a method for polymerization VDF and other monomers using manganese compound as radical inititor under different conditions. Some interesting results were presented in the manuscript. However, following points need to be considered before it could be prceeded further:

The copolymerization of vinyl acetate and butylacrylate showed norealtionships with the title of manuscript. Further, why the authors select these two monomers for copolymerization, not other monomers or homopolymerization? A scheme of the mechanism for the polymerization need to be showed in the manuscript. The mechanism is also need to be confirmed by experimental work, such as investigating the polymerization after adding radical trapter into the system etc. The structure of polymers need to be characterized using MALDI-TOF MS for confirm the initiation of moiety derived from pentacarbonylmanganese compounds. Most the figures showed in the manuscript contained only 3-4 data point. That is not-enough for the conclusions. Why some of the molecular weight data showed as ND in Table 1.

Author Response

The manuscript described a method for polymerization VDF and other monomers using manganese compound as radical initiator under different conditions. Some interesting results were presented in the manuscript. However, following points need to be considered before it could be proceeded further:

The copolymerization of vinyl acetate and butylacrylate showed no relationships with the title of manuscript. Further, why the authors select these two monomers for copolymerization, not other monomers or homopolymerization?

Answer: We disagree with the reviewer. In this manuscript we attempted the copolymerization of vinyl acetate and tert-butyl 2-trifluoromethylacrylate (MAF-TBE), not butyl acrylate. MAF-TBE is a fluoromonomer, and thus this copolymerization fits perfectly within the remit of the paper as announced in the title. Lines 420 and 421 state: “The aim of these investigations was to check the initiation capacity, using more reactive monomers”. This isolated case of copolymerization was attempted to verify the initiator capacity (radical generation) of all the manganese complexes, because the copolymerization of this specific comonomer mixture has recently been investigated in our laboratories.

A scheme of the mechanism for the polymerization need to be showed in the manuscript.

Answer: Scheme 1 shows the reaction scheme. Previous articles (references 23 and 25) have already supplied the mechanism of the formation fluoroalkyl radicals from manganese complexes. Thus, we consider that the mechanism is not necessary in this article.

The mechanism also needs to be confirmed by experimental work, such as investigating the polymerization after adding radical trapper into the system etc.

Answer: we published previous articles (references 23 and 25) that evidenced the homolytic Mn-RF bond cleavage in [Mn(RF)(CO)5] at various temperatures under saturation conditions with trapping of the generated RF radicals by excess of tris(trimethylsilyl)silane. They yielded activation parameters, ΔH and ΔS, representing close estimates of the homolytic bond dissociation thermodynamic parameters.

The structure of polymers need to be characterized using MALDI-TOF MS for confirm the initiation of moiety derived from pentacarbonylmanganese compounds.

Answer: we have not characterized these polymers using MALDI-TOF MS because they have rather high molar masses and therefore, they would not “fly” under these conditions.

Most the figures showed in the manuscript contained only 3-4 data point. That is not-enough for the conclusions.

Answer: We would have made a greater effort in the kinetic study if we had a controlled polymerization, as initially hoped. Instead, the study clearly shows that, although the manganese compounds are able to initiate the polymerization, they are not able to provide a persistent radical effect. It should also be considered that the VDF polymerization require a separate polymerization for each data point and the procedure is rather elaborate (gaseous monomer, sealed tube, high pressure, …). Therefore, we believe that that 3-4 points are few but sufficient to show a certain tendency and especially to validate the main conclusion that the polymerizations are not well controlled.

Why some of the molecular weight data showed as ND in Table 1.

Answer: Both these experiments (entries 5, 6 and 8) were carried out at the end of the project. Unfortunately, the SEC apparatus in the laboratory which enable to characterize the other PVDF samples was out of order (issue of the RI detector which could not be repaired). We did not dare using a SEC apparatus from another laboratory to be sure to keep the same conditions of injection, column, pressure, temperature and quality of solvent.

We have taken into account these answers in the revised manuscript.

Reviewer 2 Report

This work presents an interesting study on different initiators for the radical (co)polymerization of fluoromonomers. In general terms it is a well-written manuscript and it has many results that authors support in a good manner.

Some comments have been underlined in yellow, as well as some notes have been inserted within the manuscript. I encourage the authors to attend them to improve the quality of their work.

Author Response

This work presents an interesting study on different initiators for the radical (co)polymerization of fluoromonomers. In general terms it is a well-written manuscript and it has many results that authors support in a good manner.

Some comments have been underlined in yellow, as well as some notes have been inserted within the manuscript. I encourage the authors to attend them to improve the quality of their work.

“The signal of the second population at low molar mass can be attributed to oligomers arising from transfer reactions.” Is there any evidence that can support this affirmation?

Answer: we cannot assign such a small signal at retention time 22.5 min. But, as it is well-known that DMC may transfer the H atom from the methyl group, we assume that oligomers have been produced. This is supported by the 1H NMR spectra, which show the characteristic signals from of the transfer from DMC (e.g. Figure 3 signals e and f).

“Surprisingly, it should be noted that the dispersity slightly decreases with time.” Is there any idea of this behavior?

Answer: this is a good point, but unfortunately, we do not have any answer to explain this behavior.

Nevertheless, the thermogram begins at 0 °C. My guess is that it is the second scan, that is why it does not present the Tg value. IN the first scan did not appear the Tg? Why not to perform also the second scan from -50 °C to confirm this assumption?

Answer: from previous DSC analyses of PVDF, we always noted the absence of the inflexion point (corresponding to the Tg of PVDF), even starting at -100 °C, both in the first and in the second heating cycle. As explained in the experimental part, the heating cycles were started at -50 °C but the thermograms did not show any Tg. For a better presentation, we decided to only show the zone ranging between 0 and 200 °C.

We have considered all the other comments raised by the reviewer in the revised MS.

Reviewer 3 Report

In this manuscript, the authors investigated the potential OMRP of vinylidene fluoride using various Mn complexes as the initiators under thermal, visible light and UV radiation conditions. The structures of the Mn complexes were thoroughly studied in the previous report, i.e., ref 23. In this study, the authors investigated the polymerizations of vinylidene fluoride with these Mn complexes. Though the polymerizations were not well controlled, the study proved the successful initiation of the polymerization and reaching of high monomer conversion under proper reaction conditions. The manuscript was well organized and clearly presented. It is recommended to publish this report as in the present form. 

Author Response

In this manuscript, the authors investigated the potential OMRP of vinylidene fluoride using various Mn complexes as the initiators under thermal, visible light and UV radiation conditions. The structures of the Mn complexes were thoroughly studied in the previous report, i.e., ref 23. In this study, the authors investigated the polymerizations of vinylidene fluoride with these Mn complexes. Though the polymerizations were not well controlled, the study proved the successful initiation of the polymerization and reaching of high monomer conversion under proper reaction conditions. The manuscript was well organized and clearly presented. It is recommended to publish this report as in the present form.

Answer: We appreciate these nice comments.

Reviewer 4 Report

Comments:

This paper from Améduri, Poli and coworkers details the radical (co)polymerization of fluoromonomers using fluoroalkyl pentacarbonylmanganese(I) complexes. The VDF homopolymerization results depends on the nature of the RF group and on the activation method. The most of results and discussion are reasonable. Indeed the work is interesting, the paper is well prepared. The manuscript would be recommended for publication in Polymers after some minor revisions.

In Tables 1, entry 1: Why the VDF homopolymerization can't react at 50 oC and given a corresponding explanation. Generally, the molecular weights increased with polymerization time.

When the polymerization time of 24 h, the polymer molecular weight should be the largest, but the data in the table 1, entry 7 decreased and given a corresponding explanation.

Please add accurate polymerization conditions in Tables 1 and 2, such as the amount of catalyst, the amount of monomer, etc.

Author Response

This paper from Améduri, Poli and coworkers details the radical (co)polymerization of fluoromonomers using fluoroalkyl pentacarbonylmanganese(I) complexes. The VDF homopolymerization results depends on the nature of the RF group and on the activation method. The most of results and discussion are reasonable. Indeed the work is interesting, the paper is well prepared. The manuscript would be recommended for publication in Polymers after some minor revisions.

In Tables 1, entry 1: Why the VDF homopolymerization can't react at 50 °C and given a corresponding explanation.

Answer: in a previous article (reference 23), we demonstrated the high strength of the Mn-CF3 bond and thus the poor efficiency to generate radicals at low temperatures (half-life of 9·108 s). Lines 190 to 192 state: “Indeed, extrapolation of the previously determined activation parameters to 50 °C leads to a half-life of 9·108 s (ca. 29 years) for compound 1 at this temperature.[23]”

Generally, the molecular weights increased with polymerization time.

Answer: this tendency is observed in case of a controlled (or RDRP) trend. As we stated in the MS, we feel that transfer reactions from the solvent (DMC) are occurring.

When the polymerization time of 24 h, the polymer molecular weight should be the largest, but the data in the table 1, entry 7 decreased and given a corresponding explanation.

Answer: as mentioned above, we suppose that transfer reactions occur. The observed behavior is not surprising in the present case where the radical polymerization is not controlled.

Please add accurate polymerization conditions in Tables 1 and 2, such as the amount of catalyst, the amount of monomer, etc.

Answer: We have added the experimental conditions in the footnotes of Table 1 (Experimental conditions: 128 mg of [Mn(CF3)(CO)5], 1.5 g of VDF and 5 mL of DMC in glass Carius tubes). We do not understand the reviewer’s comment for Table 2 since the data are from thermal analyses. Table 4 indicates the [VDF]/[Mn(RF)(CO)5] initial molar ratio, and we have added the initial weight of VDF in its caption. In addition, we have added the amount of the catalyst in Table 5, while the [VAc]:[MAF-TBE]:[Mn] entry was already indicated.

We have considered all these answers in the revised MS.